# Impact of COVID and the Emergence of Social Emotional Learning on Education Majors

Patricia Kardambikis *[ID] and Vicki Donne

Department of Education, Robert Morris University, Moon Township, PA 15668, USA
* Correspondence: kardambikis@rmu.edu; Tel.: +1-412-397-6246

**Abstract:** In March of 2020, institutions of higher education sent their students home and moved their classroom instruction online. While this prevented the spread of COVID, it caused students to face unique challenges impacting their social emotional needs and mental health. As college students returned to campus, they had to learn to navigate a "new normal". Social emotional needs and mental health continue to be affected, which can significantly impair student's academic success and ultimately affect their future career and personal opportunities. This was particularly true for candidates enrolled in teacher preparation programs as they attempted to navigate the changing environments of both K-12 schools and higher education. Core components of teacher preparation programs are field experiences and student teaching. With the closure of schools and/or move to provide virtual instruction, clinical experiences for teacher candidates were impacted. The changing educational environments required teacher candidates to exhibit strong social emotional skills and to develop those skills in their K-12 students. A survey was conducted to examine perceptions of students majoring in education on their transition from high school to higher education and the impact of COVID on their academic performance and social and emotional well-being.

**Keywords:** COVID; teacher preparation; social emotional learning





## 1. Introduction

Mental health in higher education has been an increasing concern on college campuses made more pronounced with the COVID pandemic. These circumstances have tested the social emotional learning skills of post-secondary students, particularly those enrolled in teacher preparation programs.

### 1.1. Mental Health in Higher Education

Universities serve an essential role in supporting the health, education, and safety needs of their students (Zhai and Du 2020). Student mental health is an area of growing concern on college campuses with 95% of postsecondary school administrators noting that social, emotional, and behavioral health are significant issues at their institutions (Mistler et al. 2013; U.S. Department of Education 2021). Presidents and student affairs leaders listed student mental health as their number one concern as rising prevalence of depression, anxiety, eating disorders, and other stressors are being seen in student populations (Brown 2020). The general student population has grown more distressed, and researchers have theorized the prevalence of smartphones has produced a new generation of students who are less comfortable talking to one another resulting in students feeling lonely (Brown 2020). During their college years, students are feeling heightened pressure to succeed and in an "instant-gratification culture, shaped by social media and helicopter parenting," this leads to a lack of development in coping skills needed to navigate life on their own (Brown 2020, p. 6). According to the most recent National College Health Assessment, "nearly 50% of college students experience moderate or serious psychological distress"

(U.S. Department of Education 2021, p. 9). In fact, more than one in three students report having a mental-health disorder (Brown 2020).

These mental health issues can affect students' motivation, concentration, and social interactions, all critical factors for students to succeed in higher education (Son et al. 2020). According to a longitudinal study presented by Conley et al. (2020), the persistence of mental health problems in college students are linked to psychological distress, a decrease in academic performance as well as more global life effects decades later. Other critical components of well-being and adjustment throughout the college years include cognitive-affective strategies, such as coping strategies and social adjustments.

The provision of mental health services on college campuses is necessary to meet the needs of the current study body. The Mental Health Task Forces in Higher Education (SAMHSA 2021) described the following benefits to providing such comprehensive mental health services:

1. Reduced stigmatization of mental disorders, fewer untreated individuals, more college graduates (Community/Societal Benefits).
2. Reduced risk of student suicide and/or crises; improved student, faculty, and staff morale; better perception of institution and its commitment to student health; increased student retention rates. (Institutional Benefits).
3. Positive effect on family, friends, faculty, and staff; stronger social and familial relationship, higher work productivity. (Interpersonal Benefits).
4. Positive well-being, better concentration, higher GPA, lower likelihood of dropout. (Individual Student Outcomes) (SAMHSA 2021, p. 10).

College mental health professionals are available for identifying and serving college students' mental health needs in the following areas: (a) early identification and referral (Hunt and Eisenberg 2010); (b) programs to reduce the stigma of seeking services (Raghavan 2014); (c) efforts to raise awareness among college personnel about mental health first aid (Kitchener and Jorm 2008); and (d) campaigns to increase general mental health literacy on college campuses (Furlong et al. 2017; Kim et al. 2015). Managing the mental health crisis has seen higher education institutions shifting their focus more toward outreach and prevention, targeting some of the root causes. To provide outreach and prevention services, it is necessary to train faculty and staff members as gatekeepers and help them feel more comfortable offering empathy and support to students. Outreach programs such as offering "wellness" education can take different forms, from online programs, mindfulness related curriculum, and first year seminars (Brown 2020). First year seminars are small, formal introductory classes designed to acclimate students to college life, help them develop effective study strategies and time management skills, and integrate social emotional competencies such as balancing work and school, accessing career centers and tutoring centers, and increasing social relationships through clubs and extracurriculars (Parker et al. 2005; Wyatt and Bloemker 2013).

### 1.2. Social and Emotional Learning

A foundation in social and emotional learning is important to thrive in college. The Collaborative for Academic, Social and Emotional Learning (CASEL) defined social and emotional learning (SEL) as "the process through which all young people and adults acquire and apply the knowledge, skills, and attitudes to develop healthy identities, manage emotions and achieve personal and collective goals, feel, and show empathy for others, establish, and maintain supportive relationships, and make responsible and caring decisions" (CASEL 2022, "Fundamentals of SEL" section).

It encompasses five interrelated core social and emotional competencies: self-awareness, social awareness, self-management, relationship skills, and responsible decision making. These five competencies have been correlated to students' academic performance, positive social behaviors, reduced psychological distress, and success in college, work, family, and society (Mahoney et al. 2018). The definition of social and emotional learning has been recently expanded to include an equity lens, which states that, "through strengthening our

SEL competencies we are better able to connect across race, class, gender identity, sexual orientation learning needs and age" (Srinivasan 2019; as cited in Elmi 2020, p. 849).

SEL programs in K-12 schools are practices or intervention programs that support students in developing SEL skills with continued positive outcomes later in life (Taylor et al. 2017). Some of the demonstrated outcomes are improved high school and college attendance, social relationships, academic achievement, and high school graduate rates, and the development of resiliency skills to face challenges in life. These competencies are vital in college as higher education settings tend to present students with challenges in the form of less structure, more demands, new roles, and increased pressures than high school settings contributing to their struggles with stress, distress, and adjustment difficulties in college (Conley 2015). Social and emotional skills that are most relevant to higher education students are those that can promote their personal and interpersonal awareness and competence, and therefore help them navigate new and challenging academic, social, and emotional terrain. Promoting these competencies in K-12 and higher education, in turn, is likely to curb problems or maladjustment in emotional and social domains (Conley 2015). Research has demonstrated that SEL addresses the increased levels of mental health concerns. Students who demonstrate proficiency in SEL increase the likelihood of academic success, making healthier lifestyle choices and contributing to the good of the organization of which they are part (Reinert 2019). Social and emotional skills extend beyond the academic contexts and outcomes, examples include success in work, positive interpersonal relationships, and better mental health and overall well-being (Conley 2015).

### 1.3. The Pandemic and Mental Health in Higher Education

With the sudden outbreak of COVID, most institutes of higher education were mandated to send their students home early from the 2019–2020 academic year to prevent spread and protect students as well as surrounding communities. Schleicher (2020) reported that the COVID pandemic had a "severe impact on higher education" and that these closures not only affected learning, but safety as well. The crisis brought about questions on higher education ability to continue to provide networking and social opportunities as well as educational content (Schleicher 2020). The sudden change in students' learning environment and other circumstances caused students to face unique challenges, adversely impacting their social emotional needs and mental health (Lee et al. 2021). The loss of internships, on-campus jobs, and other opportunities also contributed to the stress and declining mental health of students. Over the course of the pandemic, researchers reported noticeable differences in behavioral and mental health with a higher number of self-reported cases of depression and anxiety around final exams (Lee et al. 2021).

Loss and trauma have become ongoing themes as students returned to post-secondary institutions after six months or more of remote and hybrid learning from the COVID-19 pandemic. No matter the graduating class returning to campus, concerns associated with the pandemic continue such as fear and worry about their own health and their loved ones, difficulty in concentrating, disruptions in sleep patterns, decreased social interactions due to physical distancing, and increased concerns about academic performance (SAMHSA 2021).

## 2. Context of the Study

### 2.1. Research Background to the Study

A unique group of US college students are the current freshman students who were not in a traditional classroom since their junior year in high school. During the height of the pandemic, the current college freshmen were totally immersed in online learning while enrolled in their junior year in high school and then, depending on their high school, either continued an online or hybrid model of instruction during their senior year of school. The freshman class of 2021 were expected to participate in the college classroom without the "traditional" instruction, interaction, and transition activities that occur with peers during their junior and senior years at high school. Now as college freshmen, reflections

and discussions acknowledge time spent alone learning online and how much they missed activities throughout their junior and senior years of high school (Indiana EDUC, NSSE 2021). According to data released through the National Survey of Student Engagement (NSSE), more than half (53%) of freshman students had substantial increases in levels of depression, hopelessness, and loneliness due to COVID-19 (Indiana EDUC, NSSE 2021). Additionally, sophomores were returning to college after spending their first year in remote or hybrid environments. College juniors had a traditional freshman year, then their college experience was disrupted when all their on-ground classes moved to a remote or hybrid format. Senior students are experiencing graduating in a post-pandemic economy (Indiana EDU, NSSE 2021). College students are faced with unique and different opportunities and stressors with developmental turning points associated with positive or negative changes in adjustment (Conley et al. 2020).

When planning for the return to campuses, mental health became a major concern given the substantial number of students experiencing psychological distress. The rapid spread of COVID-19 and social distancing measures imposed further affected the mental health of the general population, including college students (Kecojevic et al. 2020). Depression and anxiety among first-year college students increased during the pandemic (UNC News 2021). Findings also showed students' mental health struggles were associated with distanced learning and social isolation more so than other stressors such as work reduction or worries about coronavirus infecting them or their family or friends. Even a year following the COVID outbreak, researchers report that four out of five students say they continue to experience increased stress and/or anxiety (Hall 2021). Students reported that they were especially concerned with the quality of their education and issues stemming from social isolation. A Center for Disease Control and Prevention (CDC) report found one quarter of respondents ages 18–24 had contemplated suicide in the 30 days prior to completing the survey (Czeisler et al. 2020). The most recent data from the National Survey on Drug Use and Health found that millions of college-aged adults have had serious thoughts of suicide (Chessman and Taylor 2019). It also showed that millions of adolescents aged 12 to 17 have experienced suicidal ideation or attempts within the year showing that mental health concerns for those enrolling in colleges are prevalent, serious, and will not be going away.

These results indicate that college students are still finding it incredibly difficult to cope with the uncertainty brought on by the pandemic. That is because even good changes, such as being able to resume more activities, bring about uncertainty, and students are experiencing those changes across a vast spectrum. Some students are feeling hope, others are coping with loss, and some are expressing ambivalence (Hall 2021). It was reported by Zhai and Du (2020) that during the COVID pandemic, students who received counseling services on campus could no longer access these services in person during online instruction, which worsened their psychological symptoms and increased some students' risk for suicide and substance abuse. Despite growing awareness of a mental health crisis among undergraduate and graduate students, as well as faculty, there is co-occurring concern that adequate services are not being provided and/or students are not availing themselves of the services that are provided on college campuses (Marsh and Wilcoxin 2015). Researchers suggest that mental health should be a priority during the pandemic because of isolation and other life altering practices which makes it imperative for new interventions and prevention regulations to be set in place (Growe et al. 2020). The U.S. Department of Education (2021) underscored the need for urgent action and proclaimed: "We have the potential to accelerate support to meet the increased need for effective social, emotional, and behavioral practices and create a healthier path forward." (p. 6). Ramifications of COVID requires a greater investment in mental health, but it also requires the need to communicate that commitment to students. Institutions thrive when students are mentally and emotionally healthy. Institutions make mental health a priority by communicating consistently and clearly, supporting faculty and staff as they respond to

student needs, assessing and planning for sustained mental health support, and keeping equity at the forefront of all efforts (American Council on Education (ACE) 2020).

*2.2. Teacher Preparation Programs*

With the closure of schools and/or move to provide virtual instruction, college students enrolled in teacher preparation programs were greatly impacted as they attempted to navigate the changing environments of both K-12 schools and higher education. Candidates in teacher preparation programs take first year seminar courses, core academic content courses, education pedagogy courses, and work under a mentor teacher in the field. Field experiences such as observing, interviewing, and assisting educators through student teaching/clinical practice are well-established, core components of teacher preparation (Anderson and Stillman 2013). Practice teaching in a classroom allows teacher candidates the opportunity to connect theories they have learned with practical applications in the classroom (Tipton and Schmitt 2021). Under the mentorship of a cooperating teacher (district teacher) and university supervisor, student teachers gain experience cultivating a classroom community, organizing the physical space of the classroom, managing the behavior of students, implementing instructional strategies, and taking on the formal roles and responsibilities of their new profession (Choate et al. 2021; Tipton and Schmitt 2021). When K-12 schools were ordered to close during the Spring of 2020, student teaching/clinical experiences and field experiences for teacher candidates were impacted. This unprecedented disruption occurred almost overnight, leaving educational systems at all levels struggling to determine what schooling might look like for the remainder of the school year (Thompson et al. 2020). Field experiences moved from face-to-face experiences to virtual experiences. Student teaching requirements may have been reduced as well. K-12 students, educators, and preservice teachers were left feeling disoriented as their familiar educational environment was gone (Fagell 2020; Tipton and Schmitt 2021). As one researcher described the preservice student experience, "following a year of stress, uncertainty, and struggle in finding ways of reaching and teaching students, as well as personal loss and shared trauma, we are hesitant to find ourselves in similar contexts in the future". Eisenbach expounds, "this initial, visceral inclination to worry, or even panic, can be shared and understood as we stand in this moment of ambiguity and find ourselves encased in a tide we have yet to fully understand (Eisenbach 2021, p. 73).

The complex and changing educational environments required teacher candidates to exhibit strong social emotional skills and to develop those skills in their K-12 students. Teacher candidates and their mentor teachers needed to address the social-emotional needs of their students while also addressing academic learning through face-to-face instruction, online learning, and a constantly changing combination of both (Darling-Hammond and Hyler 2020). Historically research has shown a gap in teacher training and SEL (Murano et al. 2019; Schonert-Reichl and Zakrzewski 2014). Berman et al. (2018) suggested that colleges seldom fully integrate social and emotional dimensions of learning with academic instruction into programs for prospective teachers or into advanced degree programs. Schonert-Reichl and Zakrzewski (2014) recommended a three-pronged approach for integrating SEL into teacher training programs. The first prong provides pre-service teachers with SEL content encompassing what it is, the science behind it, and how to use it to structure lessons, effectively implement SEL programs, and create positive learning environments. Schonert-Reichl (2017) suggests that child development and classroom management classes are good starting points to integrate SEL content. The other two prongs emphasize the actual implementation of SEL in the classroom, starting with the mentoring of pre-service teachers during their student teaching and then continuing for at least the first two years of in-service teaching (Schonert-Reichl and Zakrzewski 2014).

## 3. The Present Study

Mental health is integral to a student's collegiate success, particularly for those enrolled in teacher education programs. The research mentioned in the above section has shown

how mental health challenges have risen for young adults throughout the pandemic. As college students learn to navigate what the "new normal" feels like, not only on campus but in society as a whole, this post COVID transition back to academia may be difficult. This study will examine key perceptions of Education major students reflecting not only on their transition from high school to higher education and the impact of academic performance and social and emotional well-being but also the impact of COVID. Therefore, the present study explored the following research questions:

1.  What campus services were sought out by teacher candidates during this year of post COVID transitions?
2.  How do teacher candidates perceive COVID has impacted their college experience?
3.  Did the teacher candidates indicate an understanding of the importance of maintaining their mental health?
4.  Was there a difference in perceptions of teacher candidates based on coursework at the high school or college level in social emotional learning?

### 3.1. Methods

The present descriptive study involved cross-sectional survey research conducted at three small, private colleges/universities in Pennsylvania. Undergraduate teacher candidates (n = 76) self-reported data on their high school and college experiences through an online questionnaire hosted in QuestionPro.

### 3.2. Instrument

A self-constructed questionnaire was developed to collect data on participant demographics, knowledge of and satisfaction with support services on campus, and perceptions of postsecondary students on their college experiences during the COVID pandemic. The entire questionnaire consisted of 19 closed response items, 4 open-ended questions, and 6 Likert scale sections (with 57 items). Experts reviewed the instrument prior to distribution to ensure content validity. To determine reliability of the instrument, Cronbach's alpha was calculated using responses on all Likert scale items, and results revealed very good internal consistency of the instrument ($\alpha = 0.90$).

### 3.3. Procedures

A contact person within the Education department at the three participating colleges/universities electronically distributed the surveys to students enrolled in their programs requesting that teacher candidates complete the survey. A reminder email was sent after two weeks. Participation was voluntary and students completing the survey were provided with the opportunity to be randomly selected to receive a $25 Amazon gift card. The survey was distributed to 252 Education teacher candidates. There were 76 completed questionnaires for a response rate of 30%.

### 3.4. Participants

Participants consisted of 76 teacher candidates from three colleges/universities in Western Pennsylvania. Of the 76 participants, 17.3% were male (*n* = 13) and 82.7% were female (*n* = 62). Participants were predominantly white, with 96.1% responding their race was White or Caucasian (*n* = 73) and 1.32% indicating they were African American or Black (*n* = 1), Asian or Pacific Islander (*n* = 1), or Hispanic (*n* = 1). Sex and race of the sample are similar to the demographics of teachers in general (U.S. Department of Education 2019). The sample represents students across years in their program with 15.79% freshmen (*n* = 12), 21.05% sophomores (*n* = 16), 28.95% juniors (*n* = 22), and 34.21% seniors (*n* = 26). Additionally, the sample represents students across various certification areas: 14.29% were seeking early childhood certification (*n* = 11), 9.09% middle level (*n* = 7), 25.97% secondary (*n* = 20), 3.90% K-12 certification (*n* = 3), 7.79% special education (*n* = 6), and 38.96% dual early childhood and special education (*n* = 30). Participants were asked whether education was their first choice of major, and 78.95% said yes (*n* = 60) while 21.05% said no (*n* = 16).

Additional demographics are presented in Table 1. It is unknown if the number of students who commute reflects living arrangements during the pandemic or if these numbers are similar to pre-COVID living arrangements.

**Table 1.** Participant Demographics.

| Characteristic | Responses | *n* | % |
| --- | --- | --- | --- |
| Enrollment Status | Full-time | 75 | 98.68% |
| | Part-time | 1 | 1.32% |
| International Student or Non-Resident Alien | Yes | 2 | 2.63% |
| | No | 74 | 97.37% |
| Transfer Student | Yes | 14 | 18.42% |
| | No | 62 | 81.58% |
| Living arrangements during the school year | Residence (commute from parent's home) | 32 | 42.11% |
| | Residence (live alone) | 2 | 2.63% |
| | Residence (house/apartment with friends/peers) | 20 | 26.32% |
| | Dormitory resident (single room) | 6 | 7.89% |
| | Dormitory resident (roommates) | 16 | 21.05% |
| First generation college student | Yes | 11 | 14.47% |
| | No | 65 | 85.52% |
| | Parents attended college | 52 | 80.00% |
| | Siblings attended college | 47 | 72.31% |
| | Both parents and siblings attended college | 34 | 52.31% |
| Employment Status While Attending College | No | 19 | 25.00% |
| | Yes | 57 | 75.00% |
| | 1–5 h | 12 | 21.05% |
| | 6–10 h | 19 | 33.33% |
| | 11–20 h | 15 | 26.32% |
| | >20 h | 11 | 19.30% |

## 4. Results

### 4.1. High School Experience

The high school experience of currently enrolled students in teacher preparation programs varied greatly. There were 13.33% who indicated their high school was located in an urban area (*n* = 10), 66.66% in a suburban area (*n* = 50), and 20.00% in a rural area (*n* = 15). Those students graduating in 2020 saw the second half of their senior year disrupted by COVID and those graduating in 2021 had a year and a half of high school experience during the pandemic. High school instructional delivery, access to postsecondary counseling, and access to college visits during the pandemic has varied by school district, state, and/or university. As such, participants were asked to describe their high school experience. In this sample, 35.14% of participants reported that their high school experience was impacted by the pandemic (*n* = 26) and 64.86% graduated prior to the pandemic (*n* = 48).

The type of instruction and attendance in high school were significantly different pre and post COVID (see Table 2). In total, when asked to describe the instruction during their senior year of high school, 65.33% of participants responded that the instruction was entirely in person (*n* = 49), 4.00% indicated entirely synchronous online (*n* = 3), 2.67% responded entirely asynchronous online (*n* = 2), and 28.00% indicated a combination of in-person and online (*n* = 21). All but one participant graduating pre-COVID had entirely in person instruction, while those graduating in 2020 and 2021 were significantly less likely to have entirely face to face instruction and more likely to have a combination of in person and online instruction ($X^2$ (6, *n* = 75) = 75.984, *p* < 0.001). Overall, respondents indicated that all day attendance was mandatory (64.47%, *n* = 49), while 7.89% indicated that logging onto the learning management system was the only requirement (*n* = 6), and for 18.42%, attendance included partial day (*n* = 14). Some participants provided narrative responses indicating that instruction was on specific days, i.e., Monday, Tuesday, and Wednesday, while others indicated that attendance varied by instructor. During the pandemic, school districts struggled with attendance requirements and what exactly it meant for students

to attend (was logging in sufficient, was instruction provided all day, part of the day, or alternate days). This is reflected in the results above. In looking at those who graduated only during the pandemic, 2020 and 2021, they were significantly less likely to be required to attend all day ($X^2$ (6, *n* = 76), 32.110, *p* < 0.001) with only 34% reporting they were required to attend all day.

**Table 2.** High School Experiences Impacted by the Pandemic.

|  |  | Graduated Prior to COVID (*n* = 48) | Graduated during Pandemic (*n* = 26) |
|---|---|---|---|
| Instruction | Entirely In Person | 97.92% | 4.00% |
|  | Entirely Synchronous Online | 0% | 8.00% |
|  | Entirely Asynchronous Online | 0% | 8.00% |
|  | Combination of In Person and Online | 2.08% | 80.00% |
|  | All day attendance was mandatory | 81.25% | 34.6% |
| Policy Toward Attendance | Logging onto the LMS only requirement | 0% | 19.23% |
|  | Attendance included partial day | 18.75% | 19.23% |
|  | Other | 0% | 26.92% |

The pandemic has brought to light the importance of the feeling of psychological, emotional, and physical safety. One survey question collected data on this important aspect of high school and asked participants to indicate if their high school offered any wellness programs. There were 72.22% who indicated that their high school did not offer wellness programs (*n* = 52). For those that indicated their high school offered a program, 9.72% offered a program in resiliency (*n* = 7), 22.22% on mindfulness (*n* = 16), 9.72% on grit (*n* = 7), and 16.67% offered social emotional learning (*n* = 12). One respondent indicated other and commented that their high school offered a Wellness Wednesday program. This was a closed response item so it is possible that respondents participated in other programs not listed, however, there was an 'Other' option where students could add programs not listed.

Additionally, participants were asked to rank the degree to which their high school prepared them academically to succeed in college. There were 32.88% who strongly agreed (*n* = 24), 35.62% who agreed (*n* = 26), 17.81% provided a neutral response (*n* = 13), 9.59% who disagreed (*n* = 7), and 4.11% who strongly disagreed (*n* = 3). The degree to which students felt their high school prepared them for college was not significantly different whether they graduated pre or during the pandemic.

*4.2. Transition to College*

To assist in the transition to college, freshmen students may attend various orientation events and take a First Year Seminar program. Teacher candidates in the sample were asked to rate the degree to which these specific programs helped them become acquainted with or adjusted to college and results are presented in Table 3. Over half of participants indicated that these programs, to a moderate or large degree, assisted with the registration process, managing college course load, and access and availability of support services such as tutoring and career services. Those students living in the dorms were more likely to report that the first-year seminar course assisted in adjusting to living on their own to a moderate or large degree ($X^2$ (16, *n* = 75) = 42.913, *p* < 0.001). There were no significant differences in perceptions of transition programs for those graduating prior to or during the pandemic.

**Table 3.** Perceptions on Orientation Events/First Year Seminar.

| Area | Not Applicable | | Not at All | | To a Small Degree | | To a Moderate Degree | | To a Large Degree | |
|---|---|---|---|---|---|---|---|---|---|---|
| | *n* | % | *n* | % | *n* | % | *n* | % | *n* | % |
| Use of technology on campus | 9 | 12.00% | 8 | 10.67% | 23 | 30.67% | 15 | 20.00% | 20 | 26.67% |
| The registration process | 7 | 9.33% | 10 | 13.33% | 19 | 25.53% | 19 | 25.53% | 20 | 26.67% |
| Adjusting to living on your own | 22 | 29.33% | 10 | 13.33% | 14 | 18.67% | 10 | 13.33% | 19 | 25.53% |
| University COVID policies and procedures | 5 | 20.00% | 12 | 16.00% | 15 | 20.00% | 17 | 22.67% | 16 | 21.33% |
| Managing college course workload | 4 | 5.33% | 9 | 12.00% | 24 | 32.00% | 15 | 20.00% | 23 | 30.67% |
| Access and availability of support services (tutoring, writing center, career services) | 5 | 6.67% | 6 | 8.00% | 22 | 29.33% | 21 | 28.00% | 21 | 28.00% |
| Access to and availability of mental health services on campus | 9 | 12.00% | 13 | 17.33% | 20 | 26.67% | 14 | 18.67% | 19 | 25.33% |

### 4.3. Academic Programs

During the pandemic, the format for instruction provided at colleges/universities also varied by institutions across the country (Schleicher 2020). To obtain information about the format of instruction for this sample, teacher candidates were asked to share information on their college schedule for the current year, fall and spring. There were 2.70% who indicated they had all online classes (*n* = 2), 56.76% indicated all in person, face to face classes (*n* = 42), and 40.54% indicated they had a mix of in person and online classes (37.84% indicated that 75% of their classes were in person and 25% were online (*n* = 28) and 2.70% have a 50% split of in person and online classes (*n* = 2), none of the respondents indicated they had 25% of classes in person and 75% online). In order to determine if the instructional mode matched candidates' preferred method of learning, participants were asked to identify their preferred learning mode. There were 2.70% of respondents who indicated they preferred online classes (*n* = 2), 70.27% preferred face to face classes (*n* = 52), 18.92% preferred hybrid classes or a combination of face-to-face and online instruction (*n* = 14), and 8.11% indicated no preference in modality. There was no correlation between the preferred learning preference and the actual course schedule modality ($X^2$ (9, *n* = 74) = 7.111, *p* = 0.626).

To determine if there were other aspects of the college academic experience impacted by the pandemic, participants were asked the degree to which they agreed with several statements regarding academics and results are presented in Table 4. Almost all participants indicated that their family supports them attending college. Results further revealed that although many students are working while attending college, they do not feel that their work negatively impacts their academics. Most respondents indicated that their study habits are effective. About 80% reported that emotional and mental health impacts their academic performance and that they understand the importance of managing stress. There were several items which collected data specifically on the impact of the pandemic on academics. Most students, 86.3%, indicated that COVID has not impacted their ability to work on academics with classmates outside of class. Conversely, 43.84% of respondents indicated that COVID has impacted their motivation to do what it takes to succeed in college and 38.88% indicated that it has impacted their attendance. In summary, findings indicate that teacher candidates understand that emotional and mental health impacts their academic performance and many teacher candidates report that COVID has impacted their academic performance in terms of attendance and motivation.

**Table 4.** Perceptions of Factors Impacting Academics.

| Statement | Disagree or Strongly Disagree | | Neutral | | Agree or Strongly Agree | |
|---|---|---|---|---|---|---|
| | *n* | % | *n* | % | *n* | % |
| I spend more time on social media now than I did pre-COVID. | 16 | 21.89% | 19 | 26.03% | 38 | 52.06% |
| My family is very supportive of my attending college. | 0 | 0% | 1 | 1.37% | 72 | 98.63% |
| I am unable to work on academics with classmates in class or outside of class due to COVID. | 63 | 86.3% | 5 | 6.85% | 5 | 6.81% |
| My way of studying for tests is effective. | 3 | 4.11% | 14 | 19.18% | 56 | 76.71% |
| COVID has not impacted my motivation to do what it takes to succeed in college. | 32 | 43.84% | 11 | 15.07% | 30 | 41.10% |
| I feel that my emotional or mental health impacts my academic performance. | 5 | 6.85% | 10 | 13.7% | 58 | 79.45% |
| My job interferes with my schoolwork. | 35 | 48.61% | 20 | 27.78% | 17 | 23.61% |
| My college experience has helped me learn to adapt to change (technology, policies, and ways of communicating). | 4 | 5.56% | 13 | 18.06% | 55 | 76.39% |
| COVID has not impacted my attendance in class. | 28 | 38.88% | 4 | 5.56% | 40 | 55.55% |
| Someone at the university contacts me if I am struggling with my classes. | 34 | 33.34% | 30 | 41.67% | 18 | 25.00% |
| I understand how my demographic characteristics (socioeconomic status, gender, age, and/or race/ethnicity) can impact my success in college. | 10 | 13.89% | 17 | 23.61% | 45 | 62.50% |
| I have thought about how my personal characteristics (such as optimism, introvert, academic abilities, values, and resiliency) can impact my success in college. | 4 | 6.07% | 7 | 9.72% | 61 | 85.23% |
| I understand the importance of managing stress. | 2 | 2.78% | 6 | 8.33% | 64 | 88.89% |

*4.4. Teacher Preparation Program and Clinical Experiences*

To obtain perceptions of the teacher preparation programs specifically, several Likert scale questions were included (see Table 5 for results). Participants indicated that the Education faculty wanted their students to succeed and did a good job of helping students to adapt to the changes brought about by COVID. There are several unique academic aspects associated with teacher preparation programs, such as field experiences, pre-student teaching, and student teaching clinical experiences. The survey included several question items to collect information specific to these experiences. Most participants indicated that COVID has impacted their participation in field experiences (70.84%, *n* = 51). Several students commented that their field experiences moved to virtual experiences and that as a result they felt less prepared for student teaching.

There were 45.95% of respondents who were currently in their pre-student or student teaching clinical experiences (*n* = 34). At the time of the survey administration, candidate participants would have been toward the end of their experience. COVID has presented unique challenges for these teacher candidates and for the schools hosting candidates. Participants had varying responses on the impact of COVID on their student teaching/clinical experiences. There were 29.17% indicating that it has not impacted their participation (*n* = 21), many were neutral on the impact (33.33%, *n* = 24), and 37.5% agreed or strongly agreed that it impacted their participation (*n* = 27). As participants conduct student teaching at various districts, the association between participants' perceptions of impact of COVID on their clinical experiences as a result of university preparation factors or school district COVID response factors could not be determined. The pandemic has presented unique challenges to the host school districts with confronting the health and spread of COVID and the resulting staffing issues (U.S. Department of Education 2022). Schools

are challenged with covering absences and vacancies and in response the state issued new guidance permitting student teachers to serve as substitute teachers (Act 91 of 2001 2001). Results revealed that only 2.94% of those in their clinical experiences were asked to substitute teach and did (*n* = 1), 14.71% were asked but declined (*n* = 5), 20.59% were asked but their university did not permit substitute teaching (*n* = 7), and 61.76% were not asked (*n* = 21).

**Table 5.** Perceptions of Components within Teacher Preparation Programs.

| Statement | Strongly Disagree | | Disagree | | Neutral | | Agree | | Strongly Agree | |
|---|---|---|---|---|---|---|---|---|---|---|
| | *n* | % | *n* | % | *n* | % | *n* | % | *n* | % |
| I believe my Education program actively encourages student engagement among students from different economic, social, racial, or ethnic backgrounds. | 0 | 0% | 4 | 5.48% | 10 | 13.70% | 28 | 38.36% | 31 | 42.47% |
| The Education instructors want me to succeed. | 1 | 1.37% | 2 | 2.74% | 9 | 12.33% | 25 | 34.25% | 36 | 49.32% |
| The Education faculty and staff have done a good job in helping me to adapt to the changes brought on by COVID. | 4 | 5.48% | 3 | 4.11% | 19 | 26.03% | 23 | 31.51% | 24 | 32.88% |
| COVID has impacted my participation in field experiences. | 2 | 2.78% | 9 | 12.50% | 10 | 13.89% | 20 | 27.78% | 31 | 43.06% |
| COVID has impacted my participation in student teaching/clinical experiences. | 2 | 2.78% | 19 | 26.39% | 24 | 33.33% | 9 | 12.50% | 18 | 25.00% |

Participants were asked if their current teacher preparation program included course work in social emotional learning. This was a closed response item and 5.41% of participants indicated they had a separate class on social emotional learning (*n* = 4), 67.57% indicated that social emotional learning was embedded or covered in several education classes (*n* = 50), 18.92% indicated they did not have this coursework (*n* = 14), and 8.11% indicated they did not know if they had this coursework (*n* = 6). Students selecting unknown indicated that they were still early in their major and were unsure how/if social emotional learning would be covered.

*4.5. Supports Offered by the University*

In addition to the support offered by the Education faculty and school districts, teacher candidates can take advantage of general university support associated with student mental health and SEL, and these may be in the form of informal or formal support systems. Several Likert scale questions were utilized to collect data on the frequency of participation in activities outside of academic programs (see Table 6). The most frequently reported informal social supports were clubs or co-curricular groups and talking or eating with friends. Universities offer various formalized student services. Likert scale questions provided an opportunity for participants to share their knowledge about these services, how often the services were used, and how satisfied they were with the services. Results are presented in Table 7. For many of the formal support services, teacher candidates were aware of the services but did not use them (career counseling, tutoring, writing centers, academic success centers, and mental health counseling). The most frequently reported services utilized were academic advising, library services, financial aid, and technology help desk. Those students who lived in the dorm with roommates were more likely to use the library weekly than those with other living arrangements ($X^2$ (12, *n* = 68) = 24.710, *p* = 0.016). The services in which participants report satisfactory services were through academic advising, computer labs, and the library.

**Table 6.** Informal Social Supports.

| Item | Never | | Occasionally | | Often | | Very Often | |
|---|---|---|---|---|---|---|---|---|
| | *n* | % | *n* | % | *n* | % | *n* | % |
| Clubs or co-curricular groups (organizations, campus publications, student government, fraternities or sororities, or club sports) | 9 | 12.33% | 25 | 34.25% | 12 | 16.44% | 27 | 36.99% |
| Eating with friends | 7 | 9.72% | 10 | 13.89% | 21 | 29.17% | 34 | 47.22% |
| Community-based service-learning activities | 22 | 30.14% | 29 | 39.73% | 13 | 17.81% | 9 | 12.33% |
| Exercise at the university recreation center | 34 | 46.58% | 21 | 28.77% | 12 | 16.44% | 6 | 8.22% |
| Ask a friend for help with a personal problem | 9 | 12.50% | 24 | 33.33% | 18 | 25.00% | 21 | 29.17% |
| Talk with a faculty member about personal concerns | 28 | 38.89% | 23 | 31.94% | 12 | 16.67% | 9 | 12.50% |
| Tal Talk with friends about social issues | 22 | 30.56% | 26 | 36.11% | 14 | 19.44% | 10 | 13.89% |
| Cultural or social event on campus | 24 | 32.88% | 25 | 34.25% | 14 | 19.18% | 10 | 13.70% |
| Intramural or club sports team | 52 | 71.23% | 12 | 16.44% | 3 | 4.11% | 6 | 8.22% |
| Talk with someone about feeling anxious or depressed | 21 | 28.77% | 25 | 34.25% | 20 | 27.40% | 7 | 9.59% |

**Table 7.** Formal Social Supports.

| Service | How Often Did You Use? | | | | How Satisfied Were You? | | | |
|---|---|---|---|---|---|---|---|---|
| | Do Not Know about | Know about but Did Not Use | Occasionally | Weekly | N/A | Not at All | Some-What | Very |
| Academic advising/planning | 4.17% | 22.22% | 61.11% | 12.50% | 1.43% | 20.00% | 40.00% | 38.57% |
| Career counsel/job placement | 25.00% | 41.67% | 25.00% | 8.33% | 42.03% | 21.74% | 18.44% | 17.39% |
| Face-to-face tutoring | 29.58% | 47.89% | 12.68% | 9.86% | 64.18% | 14.93% | 10.45% | 10.45% |
| Online tutoring | 41.43% | 45.71% | 4.29% | 8.57% | 71.64% | 10.45% | 16.18% | 7.46% |
| Writing center | 25.25% | 49.30% | 18.31% | 7.04% | 54.41% | 8.82% | 25.00% | 20.59% |
| Financial aid assistance/advising | 10.00% | 32.86% | 42.86% | 14.29% | 27.94% | 14.71% | 12.12% | 32.35% |
| Computer labs | 30.00% | 30.00% | 31.43% | 8.57% | 42.42% | 10.61% | 34.85% | 34.85% |
| Technology help desk | 25.71% | 35.71% | 30.00% | 8.57% | 40.91% | 6.06% | 19.70% | 18.18% |
| Transfer credit assistance | 36.23% | 27.54% | 27.54% | 8.7% | 45.45% | 13.64% | 11.94% | 21.21% |
| Services to students with disabilities | 33.33% | 36.23% | 18.84% | 11.59% | 62.69% | 10.45% | 21.54% | 14.93% |
| Academic success center | 20.90% | 47.76% | 20.90% | 10.45% | 46.15% | 15.38% | 19.70% | 16.92% |
| Mental Health Counselor | 26.47% | 41.18% | 17.65% | 14.71% | 50.00% | 9.09% | 19.70% | 21.21% |
| Library resources and services | 11.76% | 22.06% | 35.29% | 30.88% | 21.21% | 12.12% | 24.24% | 42.42% |
| Services for active military and veterans | 45.59% | 41.18% | 7.35% | 5.88% | 76.56% | 9.38% | 7.81% | 6.25% |

*4.6. Summary*

Teacher candidates from three private colleges/universities were surveyed on their perceptions of their high school experiences in preparation for college, their college experiences, and the impact of COVID. Most teacher candidates reported that their high school prepared them for the academics of college and indicated that their study habits were effective. Few high schools offered wellness programs to prepare students socially or emotionally. As expected, the format of instruction offered to candidates in high school were significantly different for those graduating pre-COVID and those graduating during the pandemic. Additionally, the attendance requirements were significantly different for those graduating pre-COVID than those graduating during the pandemic.

The first research question investigated the campus services sought out by teacher candidates. Students reported that the first-year seminar course assisted with the registration process, managing course loads, and access and availability of tutoring and career services. During their campus experiences, students were aware of but did not use mental health counseling. Teacher candidates did report participating in clubs and extracurricular activities. Additionally, they used the university services of academic advising, library, financial aid, and technology help desks.

An exploration of how teacher candidates perceived COVID impacted their college experience was research question 2. Many students indicated that they spend more time on social media now than pre-COVID (52%). Overall, 44% indicated COVID impacted their motivation and 39% reported it impacted their attendance. It did not, however, impact their ability to collaborate with classmates in class or outside of class. Teacher candidates indicated that COVID impacted their field experiences (70%), but fewer candidates indicated that it impacted their participation in student teaching/clinical experiences (37%). Generally, most teacher candidates indicated that faculty did a good job of helping students adapt to changes brought about by COVID.

Research question 3 examined whether teacher candidates indicated an understanding of the importance of maintaining their mental health. One question directly addressed this research question and 80% of participants indicated that emotional and mental health impacts their academic performance. Additional questions collected information on aspects of mental health, for example, 89% of participants indicated they can manage stress while 44% indicated that COVID has impacted their motivation. There were 76% of teacher candidates who agreed or strongly agreed that the college experience has helped them learn to adapt to change or become resilient. Additionally, 85% have thought about how their personal characteristics (such as optimism, introvert, academic abilities, values, and resiliency) can impact their success in college.

Finally, this study examined if there was a difference in perceptions of teacher candidates on their college experiences based on coursework at the high school or college level in social emotional learning (research question 4). Only 28% of participants indicated their high school offered any type of wellness program. There were no significant differences in perceptions based on coursework at the high school level and this may be due to the small number of participants indicating involvement in social emotional learning classes in high school. Most teacher candidates (68%) reported that social emotional learning was embedded throughout their college coursework. There were no significant differences in perceptions of the college experience based on social emotional learning coursework at the college level. The questionnaire did not assess the depth of SEL information coverage or how effectively this information was conveyed.

## 5. Discussion

The literature review, undertaken in Sections 1 and 2 above, described the growing concern of student mental health on college campuses (Brown 2020; Mistler et al. 2013; U.S. Department of Education 2021). Mental health issues can affect students' motivation, concentration, academic performance, and social interactions (Conley et al. 2020; Son et al. 2020). Education majors in the present study overwhelmingly reported that their emotional or mental health impacted their academic performance during COVID. Many participants indicated that COVID impacted their motivation and attendance. Additionally, researchers noted the negative influence of social media on mental health (Brown 2020). Students in the present study reported that they spend more time with social media now than pre-COVID, however, this study did not investigate direct links of social media, whether positive or negative, to academic performance, social interactions, or mental health.

The provision of mental health services on campus is important to meet the needs of today's college students (Brown 2020; Furlong et al. 2017; Kim et al. 2015; SAMHSA 2021). Results of the present study indicated that 44% of participants were moderately or to a large degree made aware of access to and availability of mental health services on campus

during their first-year seminar. The awareness of mental health services increased to 74% as students continued in their program. A much small number, 32%, used mental health counseling. It is unknown if lack of use was due to negative stigma or if students did not believe they needed assistance. These findings may support the concern expressed by researchers that students are not availing themselves of the mental health services that are provided on college campuses (Marsh and Wilcoxin 2015) and that colleges need to increase general mental health literacy and accessibility (Furlong et al. 2017; Kim et al. 2015).

It is important for teacher candidates to have strong SEL skills and to be able to build those skills in their K-12 students. Over half of the students in the present study reported that they had some component of social emotional learning integrated into their coursework in their university program as compared to 16.7% reporting exposure to social emotional learning offered in high school. Teacher preparation in social emotional learning may involve a dedicated course or contented embedded within other courses. Only 5% of teacher candidates reported a dedicated course on SEL. This is lower than the 13% reported by Schonert-Reichl (2017). Although social emotional learning and trauma informed care are not included among the current state teacher competencies, the state proposed adding these competencies to be embedded within coursework.

Social emotional learning has also been correlated with students' academic performance (Mahoney et al. 2018; Reinert 2019) and mental health (Conley 2015; Reinert 2019). Results of the present study indicate that teacher candidates feel they possess skills in SEL. For example, candidates indicate they are resilient 76% of teacher candidates agreed or strongly agreed that the college experience has helped them learn to adapt to change. Additionally, most teacher candidates consider how optimism and resilience can impact their success in college. Students indicated they were able to maintain supportive relationships and relied on friends, especially in social settings such as eating with friends or asking a friend for help when personal problems arise. Additionally, they indicated that they were able to achieve a successful work-school balance.

Due to COVID mitigation practices implemented in universities, many students lost internships, or practice in the field, which had a negative impact on their mental health (Lee et al. 2021). Teacher candidates in the present study indicated COVID impacted their field experiences as many of these experiences transitioned to a virtual environment. Teacher candidates reported that while completing student teaching clinical experiences, COVID did not impact their experiences as much. Ultimately these virtual field experiences could prove to have a positive impact on their overall exposure to the various educational environments they may encounter in the field of education. The most recent statistics indicate that 33% of public schools offer full-time remote instruction (U.S. Department of Education 2022). These virtual fields can provide valuable experiences to prepare teacher candidates for these remote and face to face environments.

*Limitations and Implications of the Study*

There are some limitations to the findings of the study. The 30% response rate was low. Although there is no universal acceptable response rate, the average response rate of electronic surveys was 34% (Shih and Fan 2008). However, for electronic questionnaires involving college students, researchers reported that with 50–75 respondents, response rates of 20–25% yield consistent, acceptable results (Fosnacht et al. 2017). Additionally, the respondents were predominantly white females. This is similar to the demographics of teachers in general (U.S. Department of Education 2019) and similar to survey research in higher education in general, where white female students are more likely to respond than males and non-Whites (Porter and Umbach 2006).

Findings of the present study and results of the literature review have several implications for teacher preparation programs and institutions of higher education:

- Provide professional development on mental health and SEL to university faculty, staff, and student leaders such as peer leaders or residence assistants. This may increase awareness, not just the awareness of availability of services but when and how those services may be of assistance.
- Partner with the University departments or centers for student or academic success, counseling services, and health services. This will lead to sharing of information concerning student academic success, retention rates, referrals for mental health—especially dealing with anxiety, depression, suicidal thoughts, all relating to underlying issues with COVID. Given the link between academics and mental health, the coordination of these referrals and delivery of services is important.
- Explore the integration of SEL into the curriculum. As students early in their program were not sure if SEL was integrated in the curriculum, it is important to map the integration of SEL from freshman year to senior year to ensure that SEL content and skills are addressed from theory to practice.
- Continue to integrate SEL into teacher preparation courses, specifically the child development and classroom management courses.

**Author Contributions:** Methodology, V.D. Software, V.D.; Validation, V.D.; Formal analysis, V.D and P.K.; Investigation, P.K. and V.D.; Resources, P.K.; Data curation, V.D. and P.K.; Writing—original draft preparation, P.K.; Writing—review and editing, P.K. and V.D. All authors have read and agreed to the published version of the manuscript.

**Funding:** This research received no external funding.

**Institutional Review Board Statement:** The study was conducted in accordance with the Robert Morris Institutional Review Board—IRB # 202202152812 Addendum. Additionally, IRB approval from LaRoche and Geneva.

**Informed Consent Statement:** Informed consent was obtained from all subjects involved in the study.

**Data Availability Statement:** Data is housed with the researchers.

**Conflicts of Interest:** The authors declare no conflict of interest.

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
