# Peer review of "Impact of COVID and the Emergence of Social Emotional Learning on Education Majors"

_socsci, doi:10.3390/socsci11120584_

Round 1
Reviewer 1 Report
The study is interesting and should be published provided the following concerns are duly addressed.
STRUCTURE
The overall structure of the paper is unbalanced.
The reader doesn’t not see where the research is going, it need a justification for the study, stated objectives, and signposting a plan of how the writer/s is going to demonstrate how the objectives were achieved by the study.
Section 1 is far too long and meandering – it is not sufficient to make claims after claims of the mental health challenges resulting from the pandemic for higher education student teachers and the benefits of S&L. This reads more like a loose literature review section – In fact, section 1.1 and 1.2 could be discarded, and the paper could well start with Section 1.3.
Section 2 should include the present section 3, and renumber the ensuing sections. Finally, Discussion should come before the Conclusions section. What is presently under section 4.6 should only be a short summary of the results, presented in a short paragraph at the end of the previous section. The Discussion should respond to the research questions and revisit what research has said.
See specific comments and suggestions in the attached manuscript.
STYLE
In need of editing for punctuation, typos, and conjunctions between clauses to ease the flow of reading and for the sake of clarity.

Reviewer 2 Report
Dear authors,
We appreciate the opportunity to review the manuscript ‘Impact of COVID and the Emergence of Social Emotional Learning on Education Majors.’ We believe the current research topic is important in our field and is timely mannered, considering the drastic educational and environmental changes during the COVID pandemic. However, there are critical areas for further development and in some parts, we believe the authors should redevelop the current manuscript with a more precise direction.
1. Introduction. We think sections 1.3 and 1.4 would be the major literature review highlighting the research gap between the previous findings and the current study; however, sections 1.1 and 1.2 would be a general introduction to help readers to guide toward the present research questions. So, these introductory parts would provide intensive information focusing on the current issues (i.e., problem statements). 1.3 The Pandemic and Mental Health in Higher Education. On page 3, please check the typo error: ‘COVOD’ -> ‘COVID’. In addition, please add further clarification on the operational definition of SEL and the complexities of it’s history in pre-service teacher training programs. Currently the paper presents about social and emotional outcomes of youth, but not about SEL training for pre-service or new teachers. Look to the respective work of C. Cipriano, C. Bailey, J. Brown, M. Brackett, and K. Schonert-Recihel for evidence. A short 2 paragraph section would do well towards the end of the introduction in transitioning to the present study and RQs and resolve the literature currently unattended to in the manuscript.
2. Materials and Methods. We recommend changing the subheading to ‘This Present Study’ and simply providing information for the research directions and questions. Some statements in section 2 overlapped with the following section 3, ‘Methods’; for example, the sentence “This study will examine key perceptions of …… also the impact of COVID.” was used twice on the same page 6. Please rephrase and remove the redundant statements.
3. Methods.
3.1. Instrument. The explanation of the instrument is not clear. For example, “The entire questionnaire was 28 questions (some questions had several prompts or items) and consisted of closed response items, open-ended questions, and Likert scale items.”, in this section, readers may not understand how you reported the reliability of Cronbach Alpha (α = .90). Please provide detailed information about the different types of questionnaires, for example, ‘we used XX items for measuring YY ….’.
3.2. Procedures. On page 7, the low response rate of the survey participants should be reported in the manuscript as a considerable limitation.
4. Results. Tables 2 and 3 reported all survey responses (n= 76) about course experiences but did not explicitly interpret the insightful findings as an independent study. The results were explained at the item level; thus, we believe some relevant items should be combined at a construct level to provide better interpretations of the difficulties of students’ mental health and social experiences.
This study’s results may partly show survey participants’ perceptions of the lack of social support issues in high school and university teaching during the pandemic. However, the current results did not make a tight bridge between the social needs and future directions of SEL, especially for the students in the education major (i.e., teacher candidates).
Conclusion. Considering the ‘how’ questions in the current study, the primary findings could be supported with depth-interview questions (not from the survey at the item level). Also, as a survey study, the sample size (n = 76) is small and collected as a small subgroup (e.g., due to the low response rate), which may have a limitation as an independent study. I recommend inviting some survey participants to the intensive interview with relevant open-ended questions and providing more insightful findings focusing on their social and career developmental support. Further, the sample is overwhelmingly white and female identifying, and this detail must be stated as both a limitation of the present findings as well as used to contextualize the results. This is of the utmost importance to the SEL field contemporarily.
Round 2
Reviewer 1 Report
I have reviewed Version 2 of this paper and am pleased that the authorshave considered my earlier remarks, and modified the text accordingly inparts.However, there are still two issues of concern for me: (1) thestructure - I have made some suggestions, which I summarised in acomment on page 7 of the attached manuscript; (2) the use of the word"academics" need to be clarified. Other minor comments have also beenprovided throughout the text. Please see the attached revised manuscript

Author Response
See attached document.
